# Simultaneous Substitution of Fe and Sr in Beta-Tricalcium Phosphate: Synthesis, Structural, Magnetic, Degradation, and Cell Adhesion Properties

**DOI:** 10.3390/ma15134702

**Published:** 2022-07-05

**Authors:** So-Min Kim, Kyung-Hyeon Yoo, Hyeonjin Kim, Yong-Il Kim, Seog-Young Yoon

**Affiliations:** 1Department of Materials Science and Engineering, Pusan National University, Busan 46241, Korea; sow78013@pusan.ac.kr (S.-M.K.); seweet07@pusan.ac.kr (K.-H.Y.); h.kim@pusan.ac.kr (H.K.); 2Department of Orthodontics, Dental Research Institute, Pusan National University, Yansan 50612, Korea; kimyongil@pusan.ac.kr

**Keywords:** beta-tricalcium phosphate, co-substitution, magnetic behavior, degradation

## Abstract

β-tricalcium phosphate is a promising bone graft substitute material with biocompatibility and high osteoinductivity. However, research on the ideal degradation and absorption for better clinical application remains a challenge. Now, we focus on modifying physicochemical properties and improving biological properties through essential ion co-substitution (Fe and Sr) in β-TCPs. Fe- and Sr-substituted and Fe/Sr co-substituted β-TCP were synthesized by aqueous co-precipitation with substitution levels ranging from 0.2 to 1.0 mol%. The β-TCP phase was detected by X-ray diffraction and Fourier transform infrared spectroscopy. Changes in Ca–O and P–O bond lengths of the co-substituted samples were observed through X-ray photoelectron spectroscopy. The results of VSM represent the M-H graph having a combination of diamagnetic and ferromagnetic properties. A TRIS–HCl solution immersion test showed that the degradation and resorption functions act synergistically on the surface of the co-substituted sample. Cell adhesion tests demonstrated that Fe enhances the initial adhesion and proliferation behavior of hDPSCs. The present work suggests that Fe and Sr co-substitution in β-TCP can be a candidate for promising bone graft materials in tissue engineering fields. In addition, the possibility of application of hyperthermia for cancer treatment can be expected.

## 1. Introduction

β-tricalcium phosphate (β-TCP, β-Ca_3_(PO_4_)_2_) is one of the promising bioceramic materials for bone recovery and reconstruction due to its interesting biological properties. In general, β-TCP bone substitutes are suitably applied to fill bone voids caused by surgery or disease and to treat bond defects. β-TCP is in the spotlight as a bone grafting material because it has biocompatibility and biodegradability [1]. An ideal bone substitute should undergo proper degradation and absorption rate in vivo [2,3]. The in vivo degradation mechanism and resorption behavior of calcium phosphates for bone repair and tissue engineering applications have been reported in many studies [2,4]. However, β-TCP has limitations in stimulating bone formation or inhibiting bone resorption. It is known to dissolve too quickly [5], and the rate of absorption was relatively slow compared to tissue regrowth. Recent studies have shown that residual β-TCP material surrounds regenerated bone [6,7]. So far, further improvement and modification studies on the in vitro degradation and absorption of β-TCP have not been investigated in depth [8,9]. In particular, in existing studies, it is difficult to clearly identify only the degradation characteristics because the concentration of ions contained in the solution itself, such as SBF [9,10,11], PBS [12], and α-MEM [13], is high.

The various physicochemical properties of calcium phosphate are closely related to each other and affect degradation [14]. An effective and viable approach to modifying these properties of synthetic β-TCP is to substitute essential trace elements such as Mg [15,16,17,18], Zn [19,20], Sr [18,21,22], Cu [23,24,25], Na [26,27,28], K [26,27], and Fe [29,30,31] with a lattice of β-TCP. To improve the biological performance of β-TCP, it has been proposed in a simple and effective way to perform Ca partial substitution with essential trace elements. The effectiveness and importance of this approach have been emphasized in several recent papers [27,32,33,34].

Following reports that different β-TCPs are substituted with various ions, co-substituted β-TCP has been actively investigated [18,35,36]. Since single-ion substitutions have the potential for new applications and improvements in material properties, it is reasonable to attempt multi-ion substitution for a greater synergistic effect. A recent study demonstrated osteogenic, antibacterial activity, and enhanced biological effects on cation co-substituted β-TCP [37,38,39,40,41]. The obtained results suggested potential future applications as new innovative biomaterials that significantly boost bone regeneration with acceptable defense against bacteria.

In this work, Fe and Sr were selected as co-substituted ions due to the special advantages that they can provide to biomaterials. Fe, an essential trace element in the blood and bone marrow, plays a positive role in the human body, contributing to processes such as vitamin D metabolism and collagen maturation [42,43,44,45]. Furthermore, Fe promotes the nucleation of apatite [46] and improves cell adhesion performance [35]. A recent study focused on the magnetic properties of Fe and showed its potential for hyperthermia applications for cancer treatment [47]. Scaffold designs with hyperthermia applications are being touted as suitable alternatives to radiation and chemotherapy in the treatment of osteosarcoma. Sr is an ion that follows the same physiological pathway as calcium [48,49], which improves the bone conductivity and bioactivity of β-TCP in bone tissue [50,51,52,53,54,55]. Additionally, at low concentrations, Sr promotes osteoblast proliferation and osteoclast inhibition performance, thereby enhancing bone resorption control and bone formation [50,51,52,53,54,55]. It is a widely known fact that Sr is attracting attention as a treatment for osteoporosis.

The main goal of this work is to simultaneously substitute Fe and Sr in β-TCP powders and correlate their properties with pristine β-TCP. Moreover, single-ion substitution was also studied for comparison with co-substitution: 0.2, 0.6, and 1.0 mol% of both dopants simultaneously to combine benefits of Fe and Sr. Structural change, degradation behavior, and magnetic properties were investigated. Furthermore, cell adhesion properties were studied through immersion in a culture medium.

## 2. Experiment Procedure

### 2.1. Powder Preparation

The pristine β-TCP, Fe- and Sr-substituted and Fe/Sr co-substituted β-TCP powders were synthesized by aqueous precipitation techniques. Calcium nitrate tetrahydrate (Ca(NO_3_)_2_∙4H_2_O > 98%, Junsei), iron (III) chloride hexahydrate (Cl_3_Fe∙6H_2_O > 99%, SIGMA-ALDRICH), strontium nitrate anhyrate (Sr(NO_3_)_2_ > 98%, Samchun), and diammonium hydrogen phosphate (NH_4_)_2_∙HPO_4_ > 98%, Junsei) were used as precursors for Ca, Fe, Sr, and P ions, respectively. To prepare the pristine β-TCP powder, an appropriate amount of Ca(NO_3_)_2_∙4H_2_O and (NH_4_)_2_∙HPO_4_ were dissolved in deionized water under stirring at 45 °C for 30 min. After preparing each solution, the Ca(NO_3_)_2_∙4H_2_O solution was slowly added dropwise (13 mL/min) to the (NH_4_)_2_∙HPO_4_ solution. The pH of the resulting solution was maintained at 7 by adding NH_4_OH. The solution was stirred for 2 h at 45 °C, and aged for 1 day in a water bath (40 °C) for precipitate maturation. For removal of unreacted material, the precipitated suspension was washed and filtered with 1 L of D.I. water. After filtration, the powders were dried at 80 °C for 24 h and ground to fine powders. On the other hand, Cl_3_Fe∙6H_2_O and Sr(NO_3_)_2_ were added to the calcium nitrate solution in an appropriate amount to synthesize the Fe- and Sr-substituted β-TCP powders. The amount of calcium was reduced while maintaining the (Ca + dopant)/P ratio of 1.5 corresponding to stoichiometric β-TCP. The ion concentration was adjusted based on the formula Ca_3−x_M_x_(PO_4_)_2_ (x = 0.01, 0.03, and 0.05; M = Fe/Sr, Fe, and Sr). Table 1 represents the sample name and ion concentration for synthesizing the various β-TCP powders. After the precipitation was completed, the suspension was washed and filtered to wash and remove unreacted products with primary distilled water. Then the powder was dried at 80 °C for 24 h and pulverized to a fine powder. The amount of all powders obtained through the synthetic process was in the range of 10–15 g. All powders were calcined at 800 °C for 2 h (at a heating rate of 3 °C/min and natural cooling) in the air.

### 2.2. Powder Characterization

Phase identification of the powder was performed using a powder X-ray diffractometer (XRD, X’ Pert 3, Malvern Panalytical) consisting of Cu Kα radiation generated at 40 kV and 40 mA at room temperature. The diffraction angle between 10 and 65° was scanned with a step size of 0.01° 2θ per second. Crystallite size (D) was calculated using the Debye–Scherrer relation (Equation (1)). In this case, K~0.94 is a constant shape factor, λ = 1.5406 Å is the wavelength of Cu Kα radiation, β is the broadening of full width at half-maximum (FWHM) calculated in radians, and θ is the Bragg’s diffraction angle.
(1)$$D=\frac{Κ\;\lambda }{\beta \cos\theta }$$

Elemental quantitative analysis of the synthesized powders was performed with an inductively coupled plasma atomic emission spectrophotometer (ICP-AES, JY HORIVA, ACTIVA). The sample pretreatment was carried out by dissolving 100 mg of sample powder in 100 mL of a solvent prepared by mixing 1.0 mL of HCl and 0.3 mL of HNO_3_ with 98.7 mL of triple-distilled water as in the previous study [56]. Infrared spectra of synthesized powders were obtained using Fourier transform infrared spectroscopy (FTIR, Nicolet iS50 Spectrometer, ThermoFisher Scientific, Waltham, MA, USA) with an attenuated total reflection (ATR) spectrometer. The transmission spectrum was recorded in the 400–1600 cm^−1^ region, and 32 scans were collected at a resolution of 4 cm^−1^. X-ray photoelectron spectroscopy (XPS, K-ALPHA+ XPS System, ThermoFisher Scientific, Waltham, MA, USA) was performed to analyze the elemental chemical state and surface composition of the powder sample. The experiment was performed at 12 kV, 72 W using AlKα radiation (photon energy 1486.6 eV) and XPS with a spot size diameter of 400 μm. The magnetic behavior of various synthesized powders was analyzed by a vibrating sample magnetometer (VSM, Microsense, EZ9, Lowell, MA, USA) at the temperature of 300 K and magnetic field −20 kOe to +20 kOe, as reported previously [57].

### 2.3. Degradation Test

A powder sample was manufactured as a cylindrical sample (D = 9 mm, H = 3 mm) using a mold and a press (1 tons for 1 min) for the degradation test. Sintering (at a heating rate of 3 °C/min and natural cooling) was performed at 1100 °C for 2 h. In vitro degradation of disc samples was determined by measuring weight changes after immersion in 100 mmol/L TRIS–HCl buffer (pH 7.4) for 7, 14, and 21 days. After immersion in TRIS–HCl buffer solution (pH 7.4) at a volume-to-surface ratio of 5 mL/cm^2^ [58], disc samples were incubated in a shaker (100 rpm) at 37 °C. Immersion liquids were refreshed for 2–3 days. The immersed discs were taken out, rinsed with distilled water, and kept at 40 °C for 48 h. The residual weight of the disc samples was measured and the weight change (%) was calculated as follows:(2)$$\mathrm{Weight}\;\mathrm{change}\;\left(\%\right)=\frac{W_{i}-W_{d}}{W_{i}}\times 100,\;\left(W_{i}=\mathrm{initial}\;\mathrm{weight},{\;W}_{d}=\mathrm{weight}\;\mathrm{after}\;\mathrm{degradation}\right)$$

After drying, all samples were coated with gold and observed by a field emission scanning electron microscope (FE-SEM, Tescan, Mira 3) (at 10 kV).

### 2.4. Cell Adhesion Test

Cell adhesion behavior was evaluated through morphology investigation adhered to the sample surface. For the cell adhesion test, the hDPSCs (human dental pulp stem cells) at passage 7 were used. The hDPSCs (hDPSCs; Lonza, Alpharetta, GA, USA) were purchased from Lonza (PT-5025) and cultured in Dulbecco’s modified Eagle’s medium (DMEM; Gibco, Grand Island, NY, USA) with 10% fetal bovine serum (FBS; Gibco, Grand Island, NY, USA) and 1% antibiotics (penicillin-streptomycin; Gibco, Grand Island, NY, USA). After immersed in culture medium for 1 or 3 days, 1 × 10^5^ cells were seeded on the cylindrical sample (for control, using non-degraded sample) for 1 day. All samples were washed in phosphate-buffered saline (PBS). Then 4% paraformaldehyde (PEA)/2.5% glutaraldehyde was fixed in PBS for 30 min. After washing with D.I. water, it was dehydrated in the following order: 30, 50, 70, 90, 94, and 100 vol% ethanol, each for 15 min. The ethanol-hexamethyldisilazane (HMDS, Sigma Aldrich) was used to dry the sample, and it was prepared at a concentration of 50 to 100 vol% and kept at room temperature for 15 min. After drying, all samples were coated with gold and observed by a field emission scanning electron microscope (FE-SEM, Tescan, Mira 3) (at 10 kV). Each sample was prepared in triplicate.

## 3. Result and Discussion

### 3.1. Powder Characterization

In the synthesis of multi-component compounds by co-precipitation, the deviation between the final product and the target stoichiometry of the actual chemical composition is a commonly known problem. The chemical composition of the synthesized powder was confirmed through ICP-OES element analysis. Table 2 summarizes the analysis results. The molar percentage of all substituted elements corresponds well to the nominal value. This can be explained in combination with XRD, FTIR, and XPS data, and shows that the synthesis by co-precipitation method is suitable for the fabrication of Fe/Sr co-substituted β-TCP powder with a high phase purity and a controllable composition.

Figure 1 shows the phase crystallinity of the pristine β-TCP, Fe- and Sr-substituted and co-substituted β-TCP powders. As shown in Figure 1a–c, all synthesized powders contained only the single phase of β-TCP (space group, R3c (161), ICDD = 01-072-7587) regardless of the type and number of substitution ions. This result is consistent with the previous report that secondary phase is not formed when the amount of dopant included is below the maximum substitution concentration [26]. Figure 2 represents the crystal structure of β-TCP and its different crystallographic sites to support the discussion.

Figure 1d–f shows that the main peak corresponding to the (2 1 7) plane moved with the type and number of substitution ions. As shown in Figure 1d, the main peak shifted to the left as the number of Fe ions increased. These phenomena can be explained by the variation of lattice parameters with the substitution of Fe in β-TCP. According to the previous report [13], Fe can preferentially replace Ca on Ca(4) and Ca(5) sites in the β-TCP structure as illustrated in Figure 2. When Fe is replaced in the Ca site, the diffraction peak shifts to higher 2θ values as the unit cell parameters of the structure decrease. This is due to the smaller ionic radii of Fe^2+^ (0.78 Å) and Fe^3+^ (0.65 Å) compared to Ca^2+^ (1.00 Å) decreasing the unit cell size. On the other hand, the main peak shifted to lower angle 2θ values with increasing amounts of Sr compared to the substitution of Fe. This is due to the larger ionic radius of Sr^2+^ (1.12 Å) compared to Ca^2+^ (1.00 Å) increasing the unit cell size. When Sr2+ is substituted with less than 5 at%, Ca(4) of β-TCP is preferentially occupied. This was identified through improved powder fitting showing Sr^2+^ substitution from Ca(4) to Ca(3) [59].

For Fe/Sr co-substituted β-TCP samples, the main peak did not vary significantly with the combined substitution of Fe and Sr with equal concentrations in the β-TCP structure. This means that the change in lattice parameters is average, and in terms of substitution sites, it is complementary due to the co-substitution of Fe and Sr ions.

Figure 3 shows the variation of the crystallite size for the pristine β-TCP, Fe- and Sr-substituted and co-substituted β-TCP powders. The crystallite size of the β-TCP was around 55.8 nm; on the other hand, the crystallite size for the Fe-substituted β-TCP powder was greatly decreased with increasing amounts of Fe. However, in the case of Sr, the crystallite size was slightly decreased with increased Sr compared to Fe substitution. Particularly, in the Sr5 sample, the crystallite size was increased to 56.6 nm. This slight increase in crystallite size in Sr5 is consistent with the literature [60] which suggests the crystallite size could be affected by electronegativity.

Figure 4 is an SEM image of the β-TCP and the Fe/Sr co-substituted β-TCP powders. All samples were in the form of a uniform and partially agglomerated powder. Compared with the powder particle size of TCP, the Fe-substituted samples were smaller, and the Sr-substituted samples were similar or slightly increased. Fe/Sr co-substituted samples were slightly smaller than TCP and have complementary average particle sizes. As a result, it can be seen that the ions substituted for TCP also affect the powder particle size.

Figure 5 shows FTIR spectra of the β-TCP and the Fe/Sr co-substituted β-TCP powders. All FTIR spectra show the vibrational mode characteristics of phosphate groups in the β-TCP crystal structure. The broad bands located at 1150–990 and 645–515 cm^−1^ indicate phosphate ν_3_ stretching mode and ν_4_ bending mode, respectively. Both absorption bands located at 970 and 940 cm^−1^ correspond to ν_1_ mode, and the weak absorption band centered at 432 cm^−1^ is assigned to ν_2_ mode. A particular absorption band at around 897 cm^−1^ for the (HPO_4_)^2−^ group in various samples indicated the substitution of Fe in β-TCP crystal structure [61]. These peaks were observed in Fe-substituted and Fe/Sr co-substituted samples. On the other hand, in the case of the Sr substitution, the band located at 1000–950 cm^−^^1^ shifts to a lower wavenumber compared to the pristine β-TCP. This comes from the increase of the length of Ca–O and P–O bonds because the Sr is preferentially substituted for the Ca(4) site of the β-TCP lattice [62]. Therefore, the binding force of the (PO_4_)^3−^ band corresponds to the ν_3_ stretching mode being reduced, and finally moved to a lower wavenumber. In the case of Fe/Sr co-substituted samples, the overall peak shape and position had the same tendency of the sum between Fe-substituted and Sr-substituted samples. However, in the FeSr3 sample, the peak intensity for the ν_3_ asymmetric stretching band located at 1000–1100 cm^−^^1^ was lower than the others. This is due to the decrease in the number of (PO_4_)^3−^ functional groups per unit volume.

Figure 6, Figure 7 and Figure 8 show high-resolution XPS spectra of the β-TCP and the Fe/Sr co-substituted β-TCP powders. Figure 6 shows the XPS survey graph of all samples. Figure 6 shows the O1s, Ca2p, and P2p spectra, consistent with previous reports [63,64,65], and Table 3 shows the peak positions. The difference between the Ca2p (347.34 eV) peak of the TCP sample and the Ca2p (346.95 eV) peak of the Fe3 sample is a result of the interaction between Ca^2+^, Fe^2+^, and Fe^3+^ ions [9]. The Fe2p (Fe2p_3/2_ and Fe2p_1/2_ peaks, 740−700 eV), Sr3d (143−124 eV), and Sr2p (Sr2p_3/2_ and Sr2p_1/2_ peaks, 285−255 eV) spectra of the samples are shown in Figure 7. The Fe2p peak is produced by Fe_3_O_4_. Multiple peaks occurring at the higher binding energy range of 710 eV correspond to Fe^3+^ ions, and the lower range peaks are attributed to Fe^2+^ ions [13]. However, in the case of Sr-substituted β-TCP, the peak positions corresponding to Ca2p were not varied with increase of substitution level. This is because Sr^2+^ substitutes the Ca^2+^ site in the β-TCP lattice with the same valence. In Figure 7b, the results of the substitution of Sr^2+^ ions are shown. As the Sr^2+^ is preferentially substituted for the Ca(4) site of TCP, the Ca–O and P–O lengths further increase [62], which increases the Ca2p binding energy. These results are consistent with previously reported literature [66]. On the other hand, in Sr5, the Ca2p peak moves to the lower energy again. It shows a tendency consistent with the crystallite size obtained from the XRD result and the FTIR data. Sr3d and Sr2p peaks clearly indicated the addition of Sr. In the case of the Sr3d peak, since the binding energy range overlaps with P2p [36,59], the graph line of TCP is omitted. Compared to TCP, the gradual reduction in Ca2p peak position of Fe/Sr co-substituted samples shows clear offset results. The radical decrease in P2p peak in FeSr3 and FeSr5 means that there has been a change in surface bonding force since FeSr3. From the data in Table 3, it can be expected that Ca–O and P–O lengths contracted as Ca2p and P2p peak positions decreased in Fe/Sr co-substituted samples. This means that the bonding angle of the β-TCP structure was changed as Fe and Sr were simultaneously substituted with Ca(4) sites [67].

### 3.2. Magnetic Characterization

Figure 9 shows the magnetic hysteresis (M-H) curves of various β-TCP samples and their resulted outcomes were consistent with the previous studies, in which calcium phosphates such as HAp and BCP including β-TCP have diamagnetic properties [68,69,70]. As shown in Figure 9a, all samples have diamagnetic behavior regardless of the type and number of substitution ions. After subtracting the diamagnetic background [71,72] from the original graph (Figure 9a), the magnetization curves clearly show the ferromagnetic behavior in a low magnetic field in Figure 9b. This behavior is consistent with other reports describing the magnetic behavior of diamagnetic materials, indicating that materials contain both diamagnetic and ferromagnetic properties [71,73]. The squareness ratio (M_r_/M_s_) values corresponding to Fe5, Sr5, and FeSr5 were 1.34, 0.18, and 0.32, respectively. This parameter value represents whether the domain structure of materials has single-domain structure (>0.5) or multi-domain structure (<0.5) [74,75]. These results indicate that Fe favors single-domain formation and Sr favors the formation of multi-domain structures. This proves that Fe has ferromagnetic properties as previously shown and suggests the possibility of improving the magnetic properties in the Fe/Sr co-substituted β-TCP sample.

### 3.3. Degradation Behavior

Figure 10 shows SEM images of the disc surface made for the degradation test. All samples have interconnected grains due to sintering reactions. Compared to TCP, Fe-substituted samples had surface roughness and had a smaller average grain size. On the other hand, Sr-substituted samples had a smooth surface, and the grain size was increased. Fe/Sr co-substituted samples had surface roughness and a slightly smaller grain size than TCP. Table 4 shows the density of the prepared disc samples. All samples were measured three times by the Archimedes method. Compared to TCP, the density of all samples remained somewhat constant regardless of ion substitution or sintering. As shown in Figure 10 and Table 4, modification of the physicochemical characteristics (grain size, roughness, and density) of the TCP surface by ion substitution may have a complex influence on degradation and cell adhesion [76].

Figure 11 shows the weight change after the degradation experiment conducted using TRIS–HCl buffer solution as an immersion liquid. Overall, the weight of samples decreased with immersion duration regardless of the type and number of substitution ions as shown in Figure 11a–c. In Figure 11a, the weight change of the Fe3 sample steeply decreased. This is consistent with the literature [13] in which Fe ions were shown to promote the initial degradation rate of β-TCP. However, Sr did not significantly affect the degradation of β-TCP. In the case of FeSr1, the weight change was relatively slow with immersion time compared to the TCP. This behavior could be explained by the substitution of Sr, which retards the degradation of β-TCP and, thus, stabilizes the β-TCP from the surface dissolution in TRIS–HCl solution.

Figure 12 represents the SEM images for the disc surface after immersion in the TRIS–HCl solution. Fe-substituted samples have a shape in which surface grains are exfoliated all at once. On the other hand, Sr-substituted samples maintain a relatively flat surface and have a shape as if a part of the surface is melted. The Fe/Sr co-substituted samples simultaneously have both phenomena affected by two ions as shown in Figure 12. That is, Fe promotes the decomposition behavior of β-TCP in TRIS–HCl solution. On the other hand, Sr accelerates the absorption behavior of the β-TCP surface during immersion in TRIS–HCl solution. While degradation refers to the process of physical disintegration and fragmentation, resorption refers to biological degradation that occurs along with cellular mechanisms [2]. As shown in Figure 13, the highlighted region in the yellow circle shows the state of crystal structures resembling β-TCP particles [77]. This is because β-TCP itself is dissolved, and the degraded particles are absorbed back to the surface [77,78]. Since Sr promotes the release of Ca ions from calcium phosphates [79,80,81,82], the resorption behavior is improved. Figure 14 shows a magnified SEM image of Fe/Sr co-substituted samples. As shown in Figure 14, it can be simultaneously observed the effect of degradation from Fe and resorption from Sr in Fe/Sr co-substituted samples. These results show the synergistic effect of co-substitution.

### 3.4. Cell Adhesion Behavior

The surface of the discs cultured with hDPSCs was examined by SEM to show cell adhesion and morphology, as shown in Figure 15. Cells grew in all samples. Figure 16 shows SEM images of TCP, Fe5, Sr5, and FeSr5 samples at 1k magnification. Fe5 and FeSr5 increased cell coverage on the surface more than TCP. On the other hand, TCP and Sr5 samples with relatively smooth surfaces have less cell adhesion and proliferation, and cells have grown horizontally flat. This is consistent with previous studies that stated Fe ions enhance initial cell adhesion and proliferation behavior [13,83]. In addition, it is noted that the surface roughness on the micron and submicron scale has a positive effect on the cell adhesion behavior.

## 4. Conclusions

Different compositions of Fe- and Sr-substituted and Fe/Sr co-substituted β-TCP powders were successfully synthesized by an aqueous co-precipitation route. All synthesized samples were confirmed to be single-phase including all functional groups corresponding to the β-TCP structure through XRD and FTIR analysis. Structural change indicates that the lattice parameters of the synthesized powder gradually decrease and increase according to the type and number of substitution ions. Fe preferentially displaces Ca at Ca(4) and Ca(5) sites in the β-TCP structure. Thus, substitution of Ca by Fe would lead to a decrease in the unit cell parameters of the structure and, thus, to shifts of diffraction peaks to higher 2θ values. On the other hand, Sr would preferentially be substituted at the Ca(4) site of β-TCP. The main peak shifted to lower-angle 2θ values with increasing amounts of Sr because of the larger ionic radius of Sr compared to Ca. In addition, for Fe/Sr co-substituted β-TCP samples, the main peak position did not change significantly in the β-TCP structure. This is complementary in terms of substitution site leading to a meaningful variation of lattice parameters. XPS results show that Fe/Sr co-substitution significantly contracted the Ca–O and P–O band lengths on the surface of β-TCP. All ion-substituted samples improved magnetic properties compared to the pristine β-TCP. In vitro degradation analysis demonstrated that Fe and Sr promote degradation behavior and resorption behavior on the surface of β-TCP, respectively. Therefore, the co-substituted samples had a synergistic effect, having simultaneous degradation and resorption behavior during immersion in culture medium. In addition, cell adhesion analysis revealed that the substitution of Fe causes the β-TCP sample to obtain micron/submicron-scale surface roughness and significantly improves cell adhesion and proliferation ability for β-TCP. These results suggest that Fe/Sr co-substitution for β-TCP could play a significant role in biomedical applications and could be considered an effective material for bone tissue regeneration applications through the control of degradation and resorption ability. Furthermore, Fe/Sr co-substituted β-TCP, due to more controlled magnetic properties, can be used for hyperthermia applications for cancer treatment.

## Figures and Tables

**Figure 1 materials-15-04702-f001:**
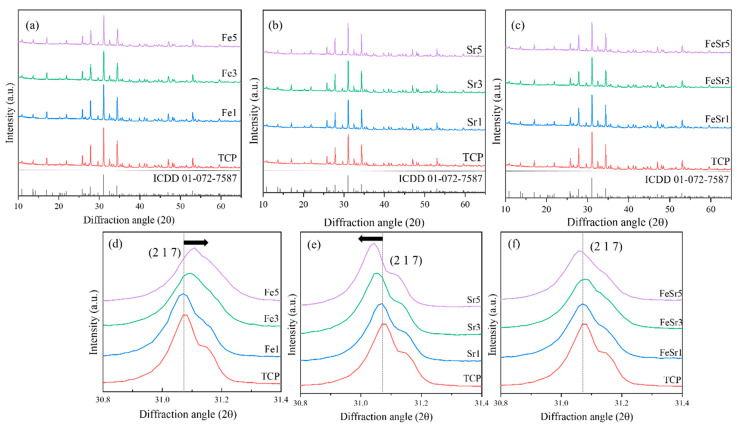
XRD patterns of various β-TCP powders with: (**a**–**c**) 2θ: 10–65, (**d**–**f**) 2θ: 30.8–31.4.

**Figure 2 materials-15-04702-f002:**
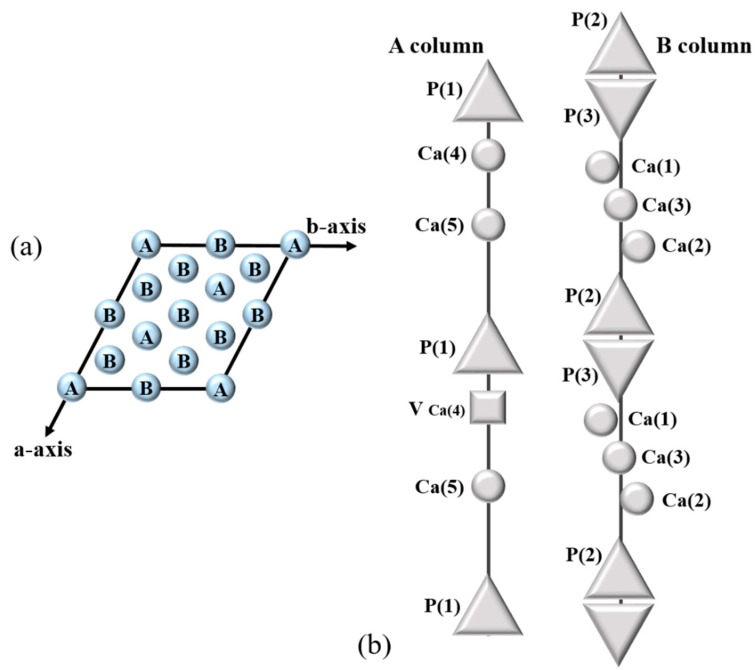
Crystal structure model of β-TCP from the view of (**a**) c-axis and (**b**) along the c-axis (modified from Yoshida et al., 2006 [26]).

**Figure 3 materials-15-04702-f003:**
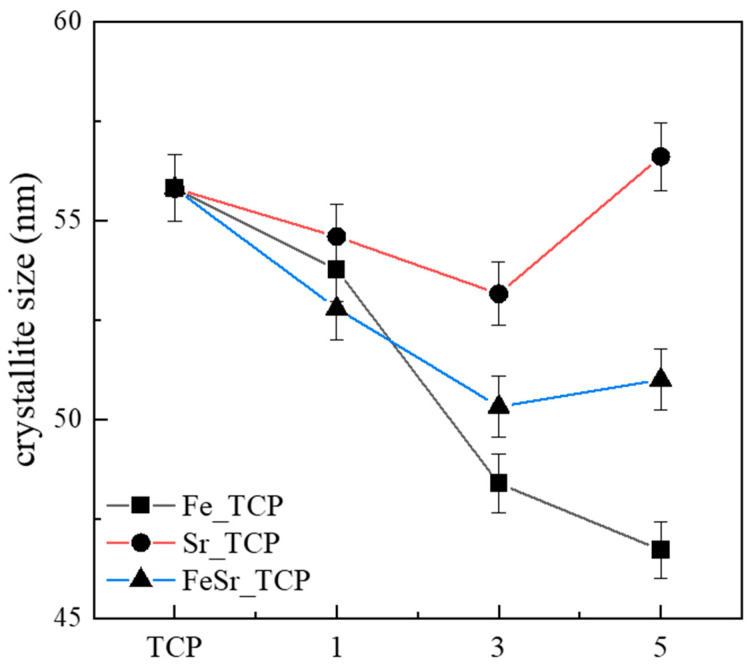
The variation of crystallite size for various β-TCP powders.

**Figure 4 materials-15-04702-f004:**
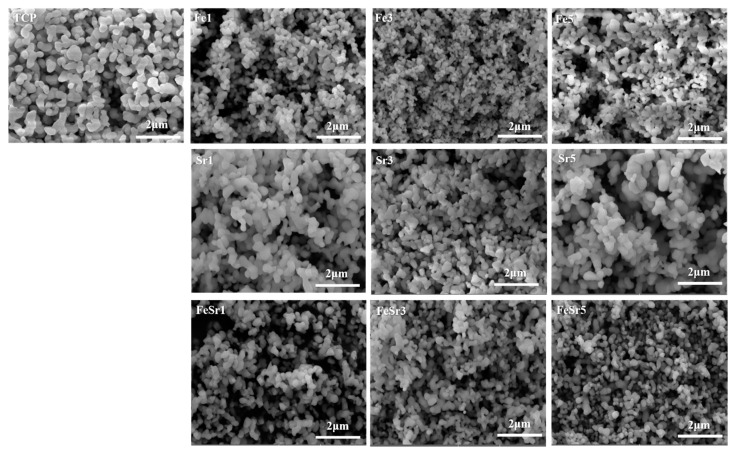
SEM images of various β-TCP powders annealed at 800 °C.

**Figure 5 materials-15-04702-f005:**
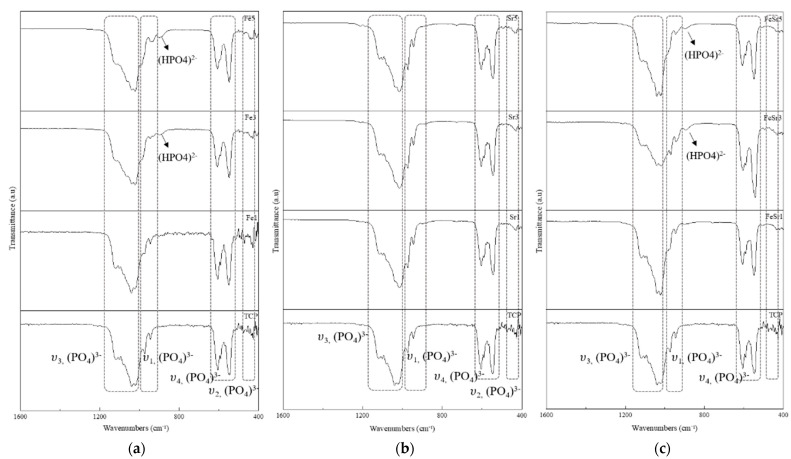
FT−IR spectra of various β-TCP powders with: (**a**) Fe−substituted samples, (**b**) Sr−substituted samples, (**c**) Fe/Sr co−substituted samples.

**Figure 6 materials-15-04702-f006:**
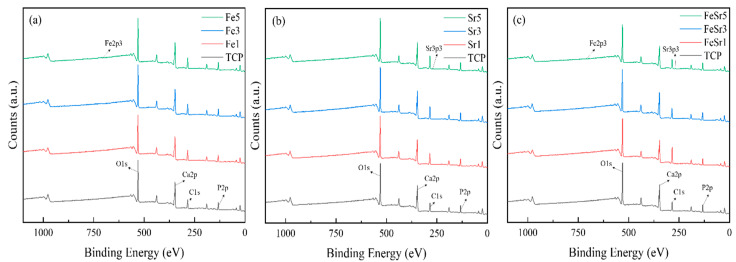
XPS spectra of various β-TCP powders with: (**a**) Fe-substituted samples, (**b**) Sr-substituted samples, (**c**) Fe/Sr co-substituted samples.

**Figure 7 materials-15-04702-f007:**
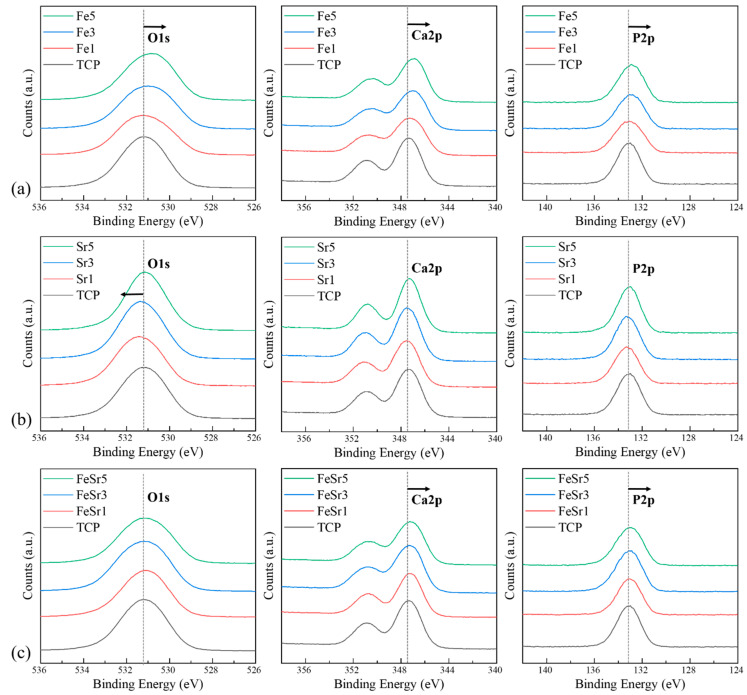
XPS spectra showing O1s, Ca2p, and P2p peaks of various β-TCP powders: (**a**) Fe-substituted samples, (**b**) Sr-substituted samples, (**c**) Fe/Sr co-substituted samples.

**Figure 8 materials-15-04702-f008:**
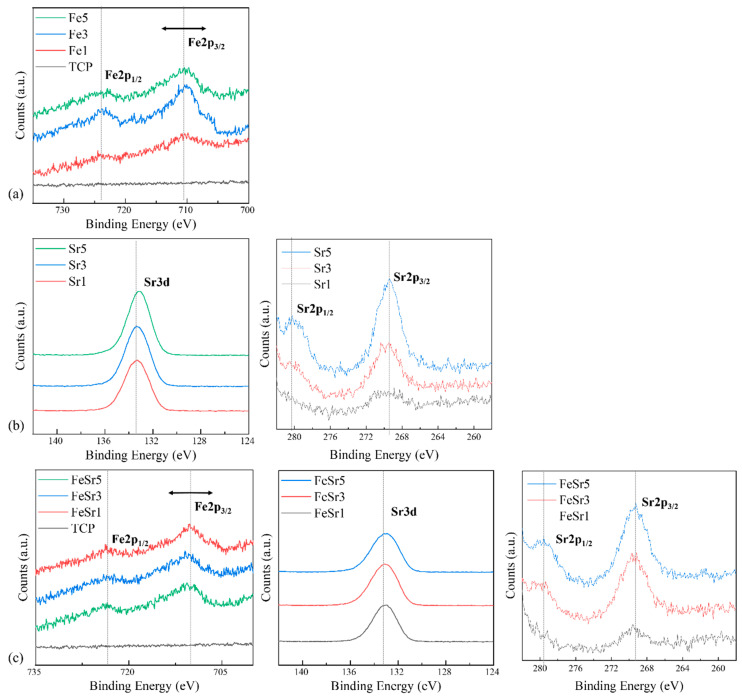
XPS spectra showing Fe2p, Sr3d, and Sr2p peaks of various β-TCP powders: (**a**) Fe-substituted samples, (**b**) Sr-substituted samples, (**c**) Fe/Sr co-substituted samples.

**Figure 9 materials-15-04702-f009:**
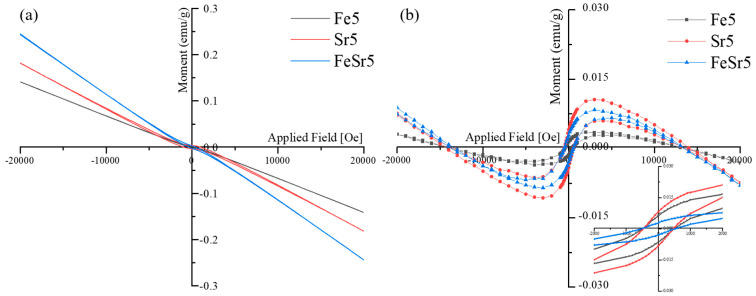
Magnetization curves of various β−TCP powders annealed at 800 °C. (**a**) origin graph, (**b**) diamagnetic behavior subtraction graph.

**Figure 10 materials-15-04702-f010:**
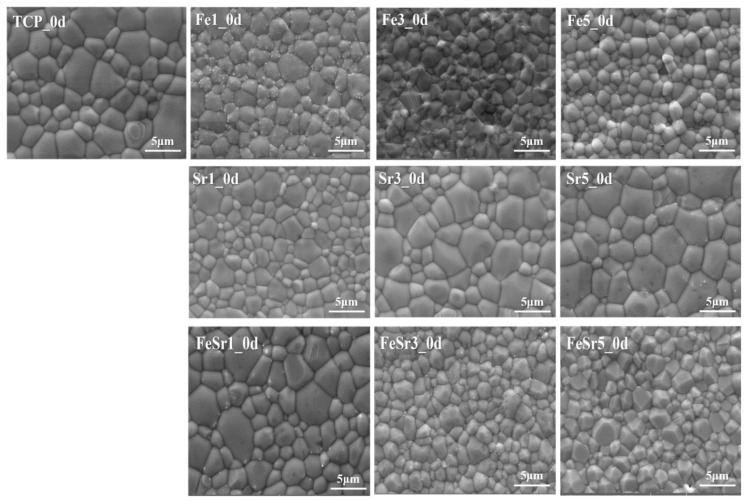
SEM images of various β-TCP discs sintered at 1100 °C.

**Figure 11 materials-15-04702-f011:**
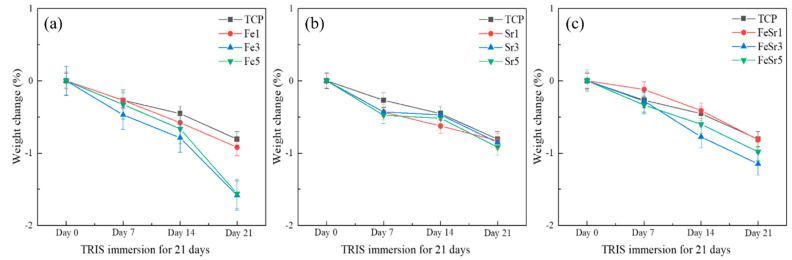
Weight change (%) after immersion in TRIS−HCl solution for 7, 14, and 21 days: (**a**) Fe−substituted samples, (**b**) Sr−substituted samples, (**c**) Fe/Sr co−substituted samples.

**Figure 12 materials-15-04702-f012:**
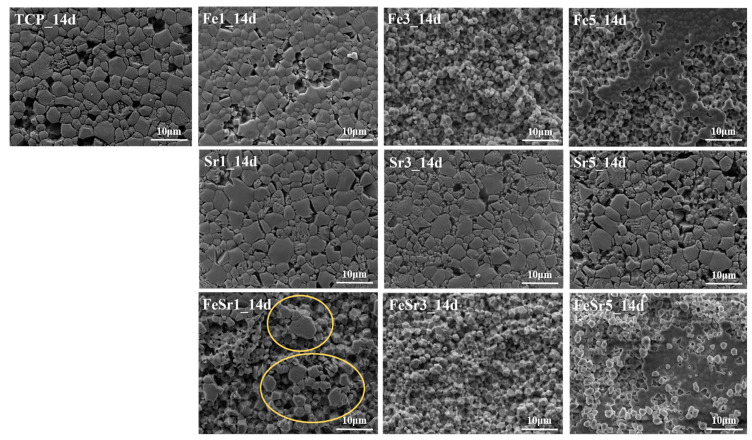
SEM image of disc surface after immersed in TRIS–HCl solution for 14 days.

**Figure 13 materials-15-04702-f013:**
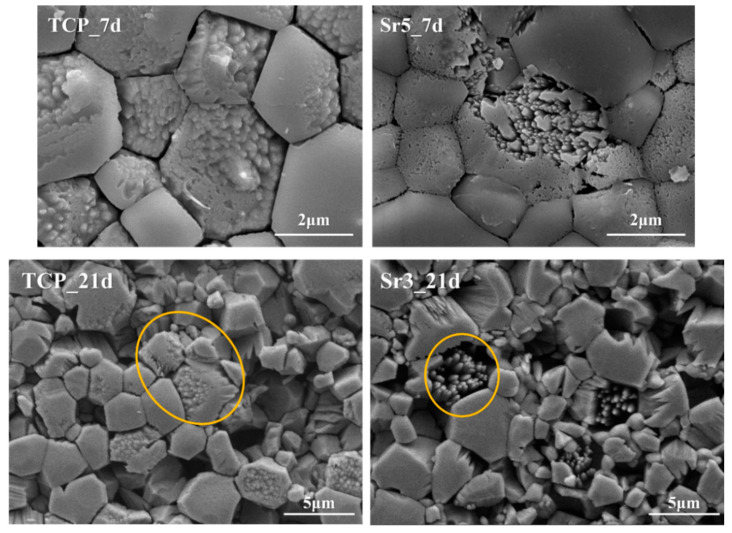
Magnified SEM images of TCP, Sr3, and Sr5 samples.

**Figure 14 materials-15-04702-f014:**
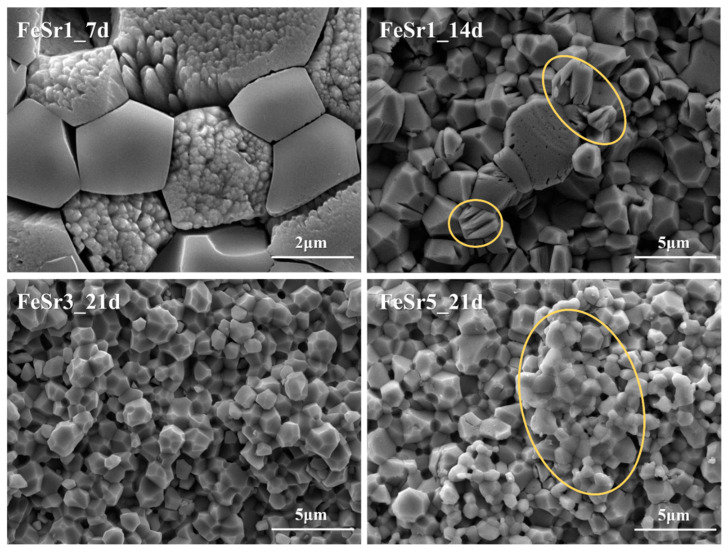
Magnified SEM images of Fe/Sr co-substituted samples.

**Figure 15 materials-15-04702-f015:**
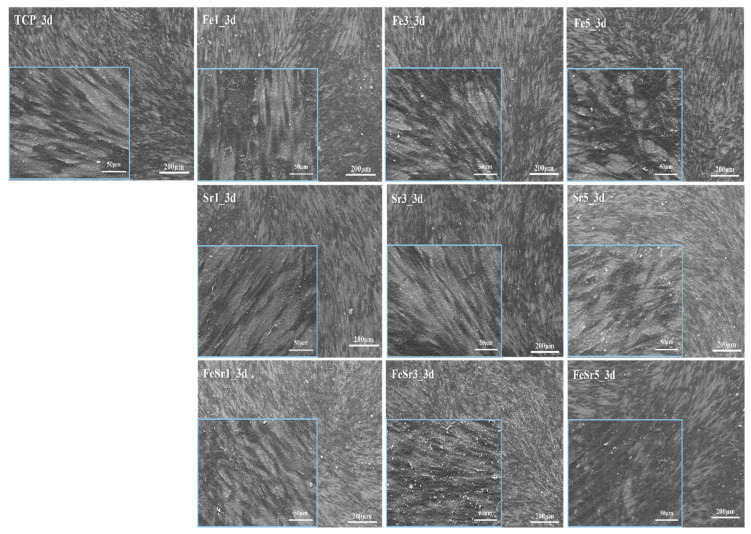
The morphology of hDPSC on the Fe- and Sr-substituted discs. (3 days).

**Figure 16 materials-15-04702-f016:**
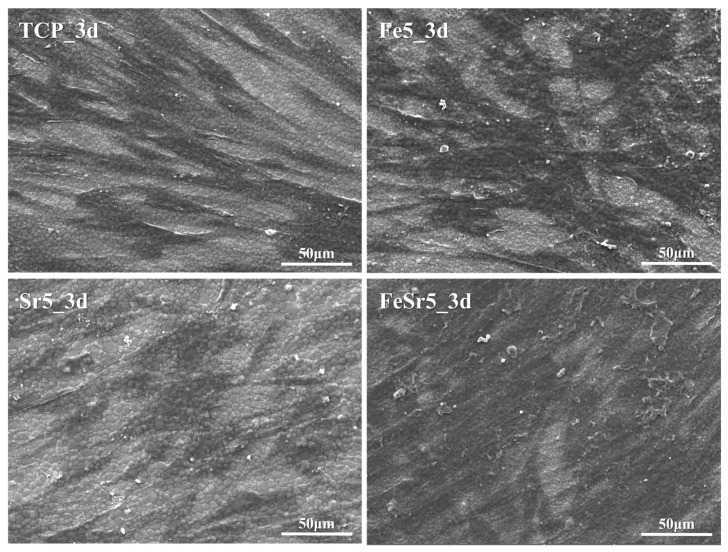
Magnified SEM images of TCP, Fe5, Sr5, and FeSr5 samples.

**Table 1 materials-15-04702-t001:** Sample notation and the concentration of chemical reagents for preparation of different β-TCP powders.

Sample Code	Fe (mol%)	Sr (mol%)
TCP	0	0
Fe1	0.2	0
Fe3	0.6	0
Fe5	1.0	0
Sr1	0	0.2
Sr3	0	0.6
Sr5	0	1.0
FeSr1	0.2	0.2
FeSr3	0.6	0.6
FeSr5	1.0	1.0

**Table 2 materials-15-04702-t002:** Chemical composition of the synthesized powders obtained by ICP-AES.

Sample	Ca (mol/kg)	P (mol/kg)	Fe (mol/kg)	Sr (mol/kg)	Ca/P	(Fe + Sr + Ca)/P
TCP	9.31	6.68	0	0	1.39	1.39
Fe1	9.08	6.44	0.03	0	1.41	1.41
Fe3	9.01	6.35	0.09	0	1.42	1.43
Fe5	8.69	6.34	0.14	0	1.37	1.39
Sr1	8.81	6.35	0	0.02	1.39	1.39
Sr3	8.96	6.38	0	0.07	1.40	1.42
Sr5	8.59	6.29	0	0.13	1.37	1.39
FeSr1	8.97	6.56	0.03	0.02	1.37	1.38
FeSr3	8.91	6.72	0.09	0.07	1.33	1.35
FeSr5	8.90	6.58	0.10	0.09	1.35	1.38

**Table 3 materials-15-04702-t003:** XPS peak position of various β-TCP powders.

Sample	Peak Position (eV)		
	O1s	Ca2p	P2p
TCP	531.18	347.34	133.11
Fe1	531.2	347.21	133.09
Fe3	530.99	347.02	132.92
Fe5	530.88	346.95	132.89
Sr1	531.4	347.51	133.28
Sr3	531.31	347.44	133.24
Sr5	531.13	347.27	133.08
FeSr1	531.11	347.22	133.05
FeSr3	531.19	347.29	129.28
FeSr5	531.12	347.19	128.88

**Table 4 materials-15-04702-t004:** Density of various β-TCP discs sintered at 1100 °C.

Sample	Density (g/cm^3^)
TCP	3.02
Fe1	3.04
Fe3	2.99
Fe5	2.78
Sr1	3.02
Sr3	3.03
Sr5	2.96
FeSr1	3.11
FeSr3	3.12
FeSr5	2.84

## Data Availability

Not applicable.

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
