# Peer review of "Simultaneous Substitution of Fe and Sr in Beta-Tricalcium Phosphate: Synthesis, Structural, Magnetic, Degradation, and Cell Adhesion Properties"

_materials, 2022, doi:10.3390/ma15134702_

Round 1

Reviewer 1 Report

Fe,Sr-doped TCP ceramics have been already reported:

10.1177/0885328218821549

However, the authors do not well explain remained problems of the previous study and what is improved in this study. Therefore, originality of the paper is quite low.

Reviewer 2 Report

The authors describe the co-substitution of bêta crystalline tricalcium phosphate (b-TCP, in my comments hereafter) with Fe and Sr to modify the properties of this material, especially in terms of biological behavior.
The mono- and co-substitution of b-TCP with Fe and Sr is classically performed by precipitation reaction (the easiest and most efficient way for a control of the synthesis) and the XRD, FTIR, ICP and XPS analyses clearly suggest that the authors have obtained the desired materials, in the defined stoichiometries.
The magnetic characterizations prove the interest of doping b-TCP with Fe to confer ferromagnetic properties to the material.
The degradation behavior in TRIS-HCl buffer solution is studied by measuring the change in weight of the samples over time and by observing the surfaces of the different samples under a scanning electron microscope.
Finally, cell adhesion tests clearly indicate an improvement of the material when b-TCP incorporates Fe and Sr in its crystal structure.
In preparation for publication of this paper, I suggest that the authors revise their manuscript with the following remarks:

- In their introduction, the authors describe the limitations of b-TCP in terms of matching the resorption rates of the material to the reformation of natural bone. They state that the in vitro properties of b-TCP have not been extensively studied. This comment should be reviewed in view of the number of existing publications on this subject, as the authors cited only a few.  Perhaps we should refer to recent review papers on the properties of calcium phosphates for bone tissue engineering?

- Substitution of b-TCP by different ions is a topic that is being studied a lot, especially due to the relative ease of synthesis (by aqueous route). Very recently, N. Somers et al. published a paper on the (co-)doping of b-TCP with Sr and Mg, which I suggest the authors read and cite (Open Ceramics, vol 7, 100168 (2021) - https://doi.org/10.1016/j.oceram.2021.100168).

- The experimental procedure for powder synthesis is then described. Would it be possible to indicate the quantities produced per synthesis?

- Regarding the characterization of the synthesized powders, several additional analyses should be added: grain size distribution, specific surface, Helium pycnometry, etc. Why did you choose to calcine the powder at 800°C for 2h, while the standards (ISO 13779-3:2008-04) suggest a treatment at 1000°C for 15h (in order to crystallize the possible residual amorphous phases)?

- It would also be interesting to know the number of samples prepared for the degradation and cells adhesion tests. The authors prepared cylindrical samples by pressing followed by sintering at 1100°C for 2h. No indication of the density of the parts is given. A prior study of the sinterability would have been logical for the characterization of a synthesized powder. In chapter 2.3 "Degradation test", the authors indicate that they collect the supernatant every 2-3 days. No characterization of this supernatant appears in the rest of the article. Elementary chemical analyses (e.g. by ICP) would have been relevant for the monitoring of degradation.

- on page 5, last sentence of the first paragraph: are the variations of the parameters of the crystal lattice significant or average? what does the term "meaning" in this sentence mean?

- Has an XRD analysis of the parts after sintering at 1100°C been performed? Are the authors sure that the b-TCP did not degrade (e.g. in the alpha phase) during the heat treatment?

- One way to study the surface changes occurring during degradation would be to measure the surface roughness (e.g., by non-contact profilometry). The microstructures observed by electron microscopy show grains of more than 10 microns, so it would be important to see the initial sizes of the powders. Fe and Sr can facilitate or inhibit sintering rates, modifying microstructures and thus cell adhesion. Physical aspects are as important as chemistry for cell behavior (I suggest the work of Sylvain Gabriele, University of Mons - Belgium, for this particular topic).

As far as English is concerned, the text is written legibly, with however some modifications to be made :

- Page 3, line 3 : "After filteration, dried at 80 o C for 24 h and ground to fine powders" to be changed in "After filtration, the powders are dried at 80°C..."

- Page 4, 3.1 Powder characterization, §2 : the second sentence is missing a verb

- Page 4, 3.1 Powder characterization, §3 : replace "These phenomena can explain with the variation of ..." by "These phenomena can be explained by the variation of..."

- Page 15-16, 4. Conclusions, last sentence : replace "Furthermore, Fe/Sr co-substituted β-TCP through more control magnetic properties can be expected to use..." by "Furthermore, Fe/Sr co-substituted b-TCP, due to more controlled magnetic properties, can be used for..."

Other spelling changes are to be corrected throughout the text.

Reviewer 3 Report

Dear authors,

We have read with interest your submitted manuscript. The overall study is well conducted and well described, with conclusions supported by the results. The research project is convincing.

The objective described in 'Introduction' section is clear, and correspond to the manuscript.

We have 1 remark:

> Intro & Methods: You used hDPSC to monitor cell adhesion. Could you briefly explain the choice of primary culture of hDPSC, and the origins of the cells. Also, the acronym DPSC must be detailed, at least once in the manuscript!

Therefore, we recommend the manuscript to be accepted fo publication, after minor revision (explain the origin of DPSC, and the interest to use such cells)

Yours faithfully,

Round 2

Reviewer 2 Report

I thank the authors for answering my various questions/suggestions in a clear and precise manner, improving, in my opinion, the quality of the article.